# Biological Characteristics of Infectious Laryngotracheitis Viruses Isolated in China

**DOI:** 10.3390/v14061200

**Published:** 2022-05-31

**Authors:** Mi Wu, Zhifei Zhang, Xin Su, Haipeng Lu, Xuesong Li, Chunxiu Yuan, Qinfang Liu, Qiaoyang Teng, Letu Geri, Zejun Li

**Affiliations:** 1College of Veterinary Medicine, Inner Mongolia Agricultural University, Hohhot 010010, China; wumiregina@outlook.com (M.W.); 15024926734@163.com (H.L.); 2Department of Avian Infectious Diseases, Shanghai Veterinary Research Institute, Chinese Academy of Agricultural Sciences, Shanghai 200241, China; nzhangzhifei@163.com (Z.Z.); suxin900220@163.com (X.S.); lixuesong@shvri.ac.cn (X.L.); yuanchx@shvri.ac.cn (C.Y.); liuqinfang@shvri.ac.cn (Q.L.); tengqy@shvri.ac.cn (Q.T.)

**Keywords:** infectious laryngotracheitis virus, biological characteristics, genome, replication, pathogenicity

## Abstract

Infectious laryngotracheitis virus (ILTV) causes severe respiratory disease in chickens and results in huge economic losses in the poultry industry worldwide. To correlate the genomic difference with the replication and pathogenicity, phenotypes of three ILTVs isolated from chickens in China from 2016 to 2018 were sequenced by high-throughput sequencing. Based on the entire genome, the isolates GD2018 and SH2017 shared 99.9% nucleotide homology, while the isolate SH2016 shared 99.7% nucleotide homology with GD2018 and SH2017, respectively. Each virus genome contained 82 ORFs encoding 77 kinds of protein, 31 of which share the same amino acid sequence in the three viruses. GD2018 and SH2017 shared 57 proteins with the same amino acid sequence, while SH2016 shared 42 and 41 proteins with the amino acid sequences of GD2018 and SH2017, respectively. SH2016 propagated efficiently in allantoic fluid and on chorioallantoic membranes (CAMs) of SPF chicken embryo eggs, while GD2018 and SH2017 proliferated well only on CAMs. GD2018 propagated most efficiently on CAMs and LMH cells among three isolates. SH2016 caused serious clinical symptoms, while GD2018 and SH2017 caused mild and moderate clinical symptoms in chickens, although the sero of the chickens infected with those three isolates were all positive for anti-ILTV antibody at 14 and 21 days after challenge. Three ILTVs with high genetic homology showed significant differences in the replication in different culture systems and the pathogenicity of chickens, providing basic materials for studying the key determinants of pathogenicity of ILTV.

## 1. Introduction

Avian infectious laryngotracheitis virus (ILTV), also named a Gallid herpesvirus 1, is a member of the Iltovirus genus, alphaherpesvirinae subfamily, and herpesviridae family [1]. ILTV causes chicken infectious laryngotracheitis (ILT), an acute, highly contagious respiratory infectious disease responsible for significant economic losses in the poultry industry worldwide. ILT was first described in the USA in 1925 [2] and first reported in Guiyang in China in the 1960s. The clinic symptoms of ILT are nasal discharge, conjunctivitis, reduced egg production, gasping, coughing, expectoration of bloody mucus, and marked dyspnea that may lead to suffocation, resulting in the mortality of ILT ranging from 5 to 70% [3]. The virulence of different ILTV isolates was various when evaluated in experimentally infected chickens [4,5,6,7]. ILTV has a linearized double-strand DNA genome of approximately 150 kb in size, which contains unique long (UL) and unique short (US) regions flanked by inverted repeat sequence (IRS) and terminal repeat sequence (TRS), and the genome normally encodes about 80 predicted viral protein open reading frames (ORFs). Using the homologous recombination knockout technique, several virulence-related factors, such as thymidine kinase (TK) [8], gJ [9], gG [10], and UL47 [11], were successfully identified in vitro or in vivo. However, the key amino acids determining the virulence of ILTV are still unclear.

The whole-genome sequences of seventy-six ILTV strains are available in the Genbank, while only a few whole-genome sequences of ILTVs isolated in China were reported before [12,13]. Limited information on ILTVs with known pathogenicity and whole-genome sequence inhibited us from understanding the molecular determinants of virulence differences of ILTVs. In this study, to correlate the genomic difference with the replication and pathogenicity phenotypes of three ILTVs (SH2016, SH2017, and GD2018) recently isolated in China, we compared their whole sequences, their replication on chicken leghorn male hepatocellular (LMH) cells, on the CAMs, and in the allantoic fluid of specific pathogen-free (SPF) chicken embryonated eggs, and their pathogenicity in chickens.

## 2. Materials and Methods

### 2.1. Cells and Viruses

LMH cells were cultured in Dulbecco’s Modified Eagle Media Nutrient Mixture F-12 (Thermo Fisher Scientific, Waltham, MA, USA) supplemented with 10% fetal bovine serum (FBS), 100 IU/mL of penicillin, and 100 μg/mL of streptomycin, and incubated at 37 °C in a humidified incubator containing 5% CO_2_. Three ILTVs, SH2016, SH2017, and GD2018, isolated from chicken farms in Shanghai (SH) and Guangdong Province (GD) in 2016, 2017, and 2018, respectively, were purified according to the methods described previously [14].

### 2.2. Sequencing

The whole genome shotgun (WGS) strategy was used to construct libraries with different inserted fragments. The virus was sequenced using a paired-end Illumina MiSeq sequencing platform by Personalbio Technology Company (Shanghai, China). A5-Miseq [15] and SPAdes [16] were used to do novo genome assembly without indexed adapter, and the contigs were constructed according to the depth of sequence assembly. The sequences with high depth were extracted and compared with the NT library on NCBI by blast [17] to pick out the viral genome sequences. MUMmer software [18] was used to analyze the synteny relationship of the above assembly results, determine the position relationship, and fill the gap between contigs; Pilon software [19] was used to correct the results to get the final viral genome sequence. Sequence alignment of the protein-coding genes was performed using diamond software. The database used for sequence alignment was NCBI NR. The protein-coding genes were compared with the protein sequence in the database by diamond blast [20].

### 2.3. Sequence Analysis

The whole genome of three ILTV strains sequenced in this study and 20 whole genomes of ILTVs available in the Genbank were analyzed using multiple alignment with fast Fourier transformation [21]. Genome-wide phylogenetic analyses were performed using the MEGA 6.0 program version 3.1 [22]. The amino acid sequences of putative proteins encoded by 82 ORFs of the three ILTVs were compared using BioEdit.

### 2.4. Proliferation Test of Three ILTVs

To study the growth properties of ILTVs, the monolayers of LMH cells, which were the optimal cell lines for ILTV amplification [23,24], were maintained in T25 culture flasks in DMEM/F12 containing 2% of FBS and infected with the ILTVs at multiple of infection (MOI) of 0.01. The culture supernatants were harvested every 24 h post-infection (hpi) for virus titration on LMH cells on 96-well plates. The assays were performed in triplicate, and controls were included as appropriate. The virus titers were calculated and expressed as median tissue culture infectious dose (TCID_50_) [25].

To study the growth properties in SPF chicken embryonated eggs, 21 10-day-old embryonated eggs were inoculated with 100 median egg infectious dose (EID_50_) of viruses in a volume of 0.2 mL onto the chorioallantoic membranes (CAMs) through the artificial air sacs, or into the allantoic cavity respectively. The CAMs of eggs inoculated on CAM or the allantoic fluid of eggs inoculated in the allantoic cavity were collected every 24 hpi. The CAMs were homogenized in 3 mL of phosphatic buffer solution (PBS), then centrifuged, and the supernatants were collected for virus titration. The virus titers were tested on LMH monolayers and calculated by Reed and Muench method.

### 2.5. Chicken Experiment

Each of nine 8-week-old SPF chickens was inoculated with 10^4.5^ EID_50_ of virus in a volume of 0.1 mL via dropping into the eyes, and negative control chickens were inoculated with PBS. At 6 days post-infection (dpi), three chickens were selected randomly from each group and euthanized. Tissue samples, including laryngotracheal, thymus glands, lungs, spleens, bursae of Fabricius, and cecum, were examined for pathology and collected for ILTV quantitative analysis by real-time polymerase chain reaction (PCR) assay. Clinical symptoms of conjunctivitis, respiratory distress, depression, and death were scored as previously described [26] until 21 days post-challenge. The clinical symptoms of individual chickens were evaluated on a scale of 0–4. No clinical symptoms were given a score of 0, mild symptoms were given a score of 1, moderate symptoms were given a score of 2, severe symptoms were given a score of 3, and death was given a score of 4. The clinical index was calculated as the total score divided by the total number of chickens, and the incidence of illness was calculated as the number of sick chickens divided by the total number of chickens.

### 2.6. Quantitative Analysis by Real-Time PCR

The primers were designed based on a conserved region of the gB gene: forward primer 5′ AGAACTCTGGTGGCAAGTATCCT 3′ and reverse primer 5′ GAACTCCTCCACGACCCTCTA 3′. The relative TaqMan probe was a 25 base pair (bp) oligonucleotide of 5′ (FAM)- CTCATCACTATCCTCCTCAACCTCC-(BHQ1) 3′. The real-time PCR was carried out in a total volume of 20 μL containing 10 μL of *2×Premix Ex Taq* (TaKaRa, China), 0.6 μL of each primer (10 μmol/ L), 0.4 μL of the probe (10 μmol/ L), 2 μL of DNA template, and 6.4 μL of deionized water. Real-time PCR was carried out on QuantStudio 5 Real-Time PCR System (Thermo Fisher Scientific). The reaction was carried out with a pre-denaturation at 95 °C for 30 s, followed by 40 cycles of denaturation at 95 °C for 5 s and annealing/elongation at 60 °C for 34 s. The fluorescent signals were measured at the end of the annealing/elongation step. Negative control was set up by substituting the DNA template with deionized water. Conditions were selected to ensure that Ct values were the lowest possible and the fluorescence acquisition curves were robust to each DNA concentration. A series of dilutions of the standard plasmid DNA were included, along with DNA samples in each run. In terms of Ct values, the quantitation data were determined using the Abs Quant/Fit Points of the LightCycler software, version 1.5.0.39 (Roche Diagnostics GmbH, Mannheim, Germany). Viral DNA was extracted from the samples of the laryngotracheal, spleen, bursa of Fabricius, lung, thymus gland, and cecum using QIAamp DNA Mini Kit, and tested and quantified by the method described above. The final concentration was calculated in copy numbers per gram of tissue samples.

### 2.7. Antibodies Detected in Chickens

Blood was collected at 0, 14, and 21 dpi from the challenged and control chickens for serologic tests. According to the manufacturer’s instructions, the specific antibodies against ILTV in serum were tested using the ELISA kit (Cellabs Pty Ltd. Sydney, NSW, Australia). Briefly, the serum diluted 100-fold with dilution buffer was added to antigen-coated plates and incubated at room temperature for one hour. Then the plates were washed three times with 300 μL of wash buffer each time and then incubated for one hour at room temperature with 50 μL of the diluted HRP-conjugated anti-chicken IgG sample solution. The plates were washed again and incubated with 50 μL of TMB for 15 min at room temperature. The reaction was stopped by adding a stop solution, and the optical density (OD) was measured at 450 nm using a BioTek epoch full wavelength microplate reader (BioTek, Winooski, VT, USA). The samples with values greater than 0.3 were considered positive for anti-ILTV antibodies. Otherwise, they were considered to be negative.

### 2.8. Statistical Analysis

Statistical analyses of virus titers were carried out using GraphPad Prism 5 (GraphPad Software Inc., San Diego, CA, USA). The mean and standard deviation (SD) were used as descriptive statistics. The student’s *t*-test was used for normally distributed variables. Virus genome copies in the tissues of chickens infected with different viruses were compared at each detection time point, and *p* < 0.05 was considered a statistically significant difference.

## 3. Results

### 3.1. Genomic Characteristics of Three ILTVs

To understand the genomic characteristics of three ILTVs, their entire genomes were completely sequenced, and the complete genome sequences were submitted to Genbank. The serial numbers of SH2016, SH2017, and GD2018 were ON415274, ON415275, and ON415276, respectively. The complete genome of SH2016, SH2017, and GD2018 contains 153,805, 152,931, and 153,855 bp of nucleotides, respectively, and each of the genomes covers four regions, the UL region, IRS, US region, and TRS. The GC contents of the genomes of SH2016, SH2017, and GD2018 were 48.16%, 48.06%, and 48.10%, respectively. The UL regions of SH2016, SH2017, and GD2018, contained 113,157 bp, 113,597 bp, and 112,915 bp of nucleotides, the US region contained 13,126 bp, 13,094 bp, and 13,192 bp of nucleotides, and both the IRS and the TRS regions contained 13,126 bp, 13,120 bp, and 13,874 bp of nucleotides, respectively. Based on the entire genome, GD2018 and SH2017 had 99.9% nucleotide homology, while SH2016 had 99.7% nucleotide homology with GD2018 and SH2017, respectively. Each virus genome contained 82 ORFs encoding 77 proteins, 31 of which share the same amino acid sequence in the three viruses (Figure A1). Those 31 proteins included capsid triplex subunit 2 encoded by UL18, deoxyribonuclease encoded by UL12, DNA packaging protein encoded by UL33, DNA packaging tegument protein encoded by UL25, DNA packaging terminase subunit 1 encoded by UL15, DNA polymerase processivity subunit encoded by UL42, envelope glycoprotein C encoded by UL44, envelope protein encoded by UL43, major capsid protein encoded by UL19, membrane protein encoded by UL45, membrane protein encoded by UL56, myristylated tegument protein encoded by UL11, nuclear egress membrane protein encoded by UL34, small capsid protein encoded by UL35, nuclear protein encoded by UL3, nuclear protein encoded by UL4, nuclear protein encoded by UL24, protein IA (ORF A), protein LORF2 encoded by UL(−1), ribonucleotide reductase subunit 2 encoded by UL40, tegument protein encoded by UL14, UL21 and UL51, tegument protein VP22 encoded by UL49, tegument serine/threonine protein kinase encoded by UL13, uracil-DNA glycosylase encoded by UL2, capsid portal protein encoded by UL6, nuclear egress lamina protein encoded by UL31, DNA packaging protein encoded by UL32, tegument protein encoded by UL37, and virion protein encoded by US2.

SH2016 was significantly different from SH2017 and GD2018 in eight proteins: envelope glycoprotein D (gD), envelope glycoprotein G (gG), envelope glycoprotein I (gI), envelope glycoprotein J (gJ), helicase-primase subunit, transcriptional regulator ICP4 (ICP4), serine/threonine-protein kinase and tegument protein VP13/14 (VP13/14). Those proteins of SH2016 shared 98.3%, 97.3%, 98.3%, 99.1%, 99.2%, 99.3%, 98.9%, and 99.3% homology with that of SH2017 and shared 98.7%, 97.3%, 98.1%, 99.1%, 99.2%, 99.2%, 98.9%, and 99.3% homology with that of GD2018, respectively. However, gG, gJ, ICP4, and VP13/14 of SH2017 shared 100% amino acid sequence homology with GD2018, suggesting that SH2017 was more closely related to GD2018 than SH2016.

### 3.2. Phylogenetics of Three ILTVs

To analyze the evolutionary relationship of different ILTV isolates, the complete genome sequences of 20 representative ILTV strains were downloaded from the Genbank in NCBI. The strain of V1-99 isolated in Australia in 1999 was placed as the root branch in the phylogenetic tree (Figure A2). SH2016 was located in the sub-lineage II with another Chinese strain, WG, in the phylogenetic tree, while SH2017 and GD2018 were located on the sub-lineage I. SH2016 was located closely with Rus/Ck/Tatarstan/2009/1643 isolated in Russia and two vaccine strain SA2 and A20 used in Australia, suggesting that this isolate might have evolved from the vaccine strains imported from Australia. SH2017 and GD2018 had 99.9% nucleotide homology, while SH2016 had 99.7% nucleotide homology with SH2017 and GD2018, suggesting that SH2016 evolved independently from SH2017 and GD2018 in China. SH2017 was closely related to two vaccine strains, LT Blen and Laryngo vac used in United States of America (U.S.), suggesting that this isolate might have evolved from the vaccine strains imported from USA.

### 3.3. Proliferation of Three ILTVs

To compare the proliferation efficiency of three ILTVs, the viruses were cultured on LMH cells, on the CAM of SPF chicken embryonated eggs, or in the allantoic fluid of SPF chicken embryonated eggs. The viruses were inoculated to infect monolayer LMH cells in T25 culture flasks at MOI of 0.01, and the culture supernatants were collected every 24 h post-infection for virus titration. GD2018 was detectable at 24 h post-infection on LMH cells on 96-well plates, while SH2016 and SH2017 were not detectable until 48 h post-infection (Figure A3A). At all the testing time points, the titers of GD2018 were significantly higher than that of the other two isolates. Although the titers of SH2016 were higher than that of SH2017 at 48 h, the titers of the two viruses tend to be the same at 96 and 120 h (Figure A3A).

To compare the replication on CAMs, each of 100 EID_50_ viruses was inoculated on the CAMs through artificial air sacs of SPF chicken embryonated eggs. Three CAMs infected with each virus were collected every 24 h and homogenized to prepare the supernatants for virus titration. SH2016 and GD2018 were detectable on the CAMs at 48 hpi, while SH2017 was not detectable until 72 hpi (Figure A3B). The titers of GD2018 were significantly higher than that of the other two isolates at 72, 96, and 168 hpi. When compared to the virus titers in pairs, the titers of GD2018 showed significantly higher than that of SH2016 at all the testing time points since 72 hpi except 144 hpi, and significantly higher than that of SH2017 at 24, 72, 96, 144, and 168 hpi. The titers of SH2016 did not show a significant difference from that of SH2016 since 48 hpi (Figure A3B). To compare the replication in allantoic fluid, 100 EID_50_ of ILTVs were inoculated into the allantoic cavity of SPF chicken embryonated eggs. Compared to CAMs, replication capabilities of SH2017 and GD2018 were poor in allantoic fluid, and SH2017 could be detectable with low titers on the testing time of 120 and 168 hpi, while GD2018 could not be detectable until 168 hpi. Although the replication ability of SH2016 in allantoic fluid was the best among three isolates, it could not be detected until 72 hpi and reached the highest titers until 168 hpi, suggesting the SH2016 multiplies more slowly in allantoic fluid than on CAMs of chicken embryonated eggs (Figure A3C).

### 3.4. Pathogenicity of ILTVs in Chickens

To test the pathogenicity of ILTVs, each 8-week-old SPF chicken was challenged by dropping 10^4.5^ EID_50_ of viruses into the eyes. The clinical symptoms of two of nine chickens infected with SH2016 were conjunctivitis or excessive lacrimation at 4 days after the challenge. All the chickens infected with SH2016 showed clinical symptoms, and five of nine chickens showed serious respiratory symptoms, distress, and depression at 6 days post-challenge. Four of nine chickens infected with SH2017 or GD2018 showed clinical symptoms at 6 days post-challenge. Two chickens infected with SH2017 showed serious respiratory distress and depression, while the clinical symptoms of the four chickens infected with GD2018 were mild or moderate. The incidence of SH2016 was 100%, and that of SH2017 and GD2018 was 44.44%, respectively. The chickens infected with SH2016 and SH2017 recovered within 10 days after the challenge, while those infected with GD2018 recovered within 8 days. The clinical symptoms peaked at 6 days after challenge with SH2016 and 7 days after challenge with SH2017 and GD2018. The peak clinical indexes of SH2016, SH2017, and GD2018 were 2.2, 0.8, and 0.8, respectively, which indicated that SH2016 was a high pathogenic strain, and SH2017 and GD2018 were low pathogenic strains (Figure A4).

To test the pathological changes of chickens after being infected with ILTVs, 3 chickens in each group were euthanized, and their organs were observed by gross anatomy method at 6 days post-challenge. Compared with the PBS control group, 3 chickens infected with SH2016 and one infected with SH2017 showed hemorrhagic tracheitis with blood clots (Figure A5). The laryngotracheal lesions of the other 2 chickens infected with SH2017 and the 3 chickens infected with GD2018 were unclear. No pathological changes were found in other organs of the chickens infected with different ILTVs. The virus in different tissues was tested and quantified by real-time PCR. The virus was detectable in the laryngotracheal tissues of 3 chickens infected with SH2016 and SH2017 and 1 infected with GD2018. The virus was not detectable in the tissues of the spleen, bursa of Fabricius, lung, thymus gland, and cecum. The virus genome copy numbers in the laryngotracheal tissues of SH2016- and SH2017-infected chickens were 7.24 ± 1.36 log10 per gram and 6.16 ± 1.22 log10 per gram, respectively, and that of the only one positive chicken infected with GD2018 was 5.95 log10 per gram.

### 3.5. Immune Response of Chickens Challenged with ILTVs

To detect the immune response of chickens infected with different ILTVs, the sero were collected at 0, 14, and 21 days post-challenge to measure antibodies by ELISA assay. The sero collected at 0 days post-challenge were negative for anti-ILTV antibody, and their OD_450_ values were less than 0.3. All of the sero of chickens infected with ILTVs converted to be positive to anti-ILTV antibodies at 14 and 21 days post-challenge (Figure A6), indicating that the three ILTV isolates could replicate well in all chickens infected by dropping into eyes.

## 4. Discussion

Infectious laryngotracheitis virus causes respiratory disease in chickens, resulting in great economic losses in the livestock and poultry industries [3]. ILTV is a double-stranded DNA virus with a highly conserved genome sequence [27]. The whole-genome sequence of 3 ILTV isolates was determined through high-throughput sequencing. The genome consists of four parts: UL, US, IRS, and TRS, the same as the IRS but has a reverse direction. The genome organizational structures are consistent with that reported previously [28,29]. In the phylogenetic tree, SH2016 is located in a different sub-lineage from SH2017 and GD2018, suggesting a large evolutionary distance between SH2016 and the other 2 viruses. Both SH2016 and strain WG isolated in China [12] located on the sub-lineage II suggest that those viruses shared the same ancestor. The attenuated vaccine strain of ILTV is easy to transmit and becomes more virulent in chicken flocks [30]. SH2017 and GD2018 were closely related to two vaccine strains, strain LT Blen and strain Laryngo vac, indicating that those viruses’ origin might be due to the vaccines introduced from the U.S. and used in China.

In this study, three ILTV isolates had high genome homology but revealed significant variances in the replication in LMH cells and embryonated chicken eggs and the pathogenicity in chickens, suggesting those ILTVs are excellent material for exploring the molecular determinants of ILTV replication and pathogenicity. The biological characteristics of the virus are closely related to its proteins [31,32,33], while three ILTVs shared 31 viral proteins with the same amino acid sequence, suggesting that those proteins are not related to the difference in the virus replication and pathogenicity. Among those 31 proteins, gC is included and can be excluded from a virus virulence determinant. SH2016 was significantly different from SH2017 and GD2018 in eight proteins of gD, gG, gI, gJ, helicase-primase subunit, ICP4, serine/threonine-protein kinase US3, and VP13/14. Among those eight proteins, gD, gG, gI, and gJ were glycoproteins that were thought to be important for ILTV replication and eliciting humoral and cell-mediated immune responses in the host [29]. Previous studies have found that gD binds to the susceptible host cell receptor [34,35], gG plays an immunomodulatory role in the chicken immune response [10,36], gI plays a role in virus cell-to-cell spread by forming a heterodimer with gE [37], and gJ plays a role in virus release [38]. Whether the amino acid mutation in gD, gG, gI, and gJ of ILTVs leads to the replication and pathogenicity differences needs further study. In addition to glycoproteins, the other proteins might also lead to differences in replication and pathogenicity. It was reported that the helicase-primase subunit, involved in the DNA replication [39], ICP4 initiating the transcription of early and late genes [40], serine/threonine-protein kinase US3 involved in viral replication, apoptosis resistance, and cell to cell spread in alphaherpesvirus [41], and VP13/14 related to virulence in chickens [11] were also worthy of further study to find out the cause of the differences between high and low pathogenic ILTVs.

In this study, the clinical signs observed in 8-week-old SPF chickens challenged with SH2016 through the eye-drop route were typical symptoms, such as inactivity, anorexia, coughing, moist rales, open-mouthed breathing, and high-pitched squawk. Those observations were consistent with earlier reports [42]. In contrast, SH2017 and GD2018 caused mild or moderate clinical symptoms in chickens. GD2018 caused a relatively short course of the disease and very mild pathogenic lesions in the larynx and trachea. In addition, GD2018 can efficiently stimulate antibodies against ILTV in all of the chickens challenged, suggesting this mild pathogenic virus has the potential to develop as a vaccine after further attenuation. Some ILTV isolates have wide tissue tropism, and they can be detected in the kidney tissue [43], the thymus tissue [44], and even in the brain and cecum [45]. However, three ILTVs recently isolated in China are detectable only in the larynx and trachea. Different ILTV isolates have different susceptibility to tissues and organs, which may be related to the difference between viral proteins and specific receptors of different tissues [46].

In summary, the whole genome of three ILTV isolates was sequenced, and their differences were compared. Although the entire genomes share more than 99.7% of nucleotide homologies, a significant difference in replication culture systems and pathogenicity in chickens were found in three ILTVs, which may result from the mutation of amino acids in gD, gG, gI, gJ, helicase-primase subunit, ICP4, serine/threonine-protein kinase US3 and VP13/14. We hypothesize that mutant sites in the above proteins affect growth characteristics and virulence and pathogenicity among ILTV by affecting protein function and will focus on mutant amino acids in the above proteins associated with virulence and pathogenicity in the next study. This study provided the basic information for the studies of key determinants of ILTV’s replication and pathogenicity.

## Data Availability

The data presented in this study are available on request from the corresponding author.

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
