# Peer review of "Biological Characteristics of Infectious Laryngotracheitis Viruses Isolated in China"

_viruses, 2022, doi:10.3390/v14061200_

Round 1
Reviewer 1 Report
Abstract – the goal of characterizing the genomes of the selected viruses and correlating specific differences with virulence/pathogenicity phenotypes is very good and should be clearly stated in the abstract as well as the introduction.
Line 61 – more detailed description of the LMH cells would be useful (origin, reasons for use of this cell line?)
Line 200 – phylogenetics results are very interesting – I would recommend emphasizing this.
Line 318 – avoid the term “show” or “showed”. It is not necessary. Better option suggested:
“…The clinical signs observed in chickens challenged with SH2016 were…..” These observations were consistent with earlier reports [42].
I did not mark this in all sites, but avoiding these terms improves readability.
Line 331 – In the summary, what could possibly account for the observed differences? What is the authors’ new hypothesis?
This was a good study and a pleasure to review.
Reviewer 2 Report
The authors compared the genome-wide sequences of three ILTV's isolated in China, their replication in chicken liver cells, in the embryonic CAMs, and in the allantoic fluid of SPE embryonated eggs. In addition, the authors studied the pathogenicity of these three strains in chickens. The authors used appropriate methods, their results are well presented and the discussion is clear and covers the results properly.
